# Lithium Niobate MEMS Antisymmetric Lamb Wave Resonators with Support Structures

**DOI:** 10.3390/mi15020195

**Published:** 2024-01-27

**Authors:** Yi Zhang, Yang Jiang, Chuying Tang, Chenkai Deng, Fangzhou Du, Jiaqi He, Qiaoyu Hu, Qing Wang, Hongyu Yu, Zhongrui Wang

**Affiliations:** 1Faculty of Engineering, The University of Hong Kong, Hong Kong 999077, China; jiangeee@connect.hku.hk; 2School of Microelectronics, Southern University of Science and Technology, Shenzhen 518055, China; 12049024@mail.sustech.edu.cn (C.T.); 12149033@mail.sustech.edu.cn (C.D.); 11811803@mail.sustech.edu.cn (F.D.); hjq447052447@163.com (J.H.); 12231180@mail.sustech.edu.cn (Q.H.); wangq7@sustech.edu.cn (Q.W.); yuhy@sustech.edu.cn (H.Y.)

**Keywords:** lithium niobate, acoustic resonators, A1 mode, spurious modes, power capacity

## Abstract

The piezoelectric thin film composed of single-crystal lithium niobate (LiNbO_3_) exhibits a remarkably high electromechanical coupling coefficient and minimal intrinsic losses, making it an optimal material for fabricating bulk acoustic wave resonators. However, contemporary first-order antisymmetric (A1) Lamb mode resonators based on LiNbO_3_ thin films face specific challenges, such as inadequate mechanical stability, limited power capacity, and the presence of multiple spurious modes, which restrict their applicability in a broader context. In this paper, we present an innovative design for A1 Lamb mode resonators that incorporates a support-pillar structure. Integration of support pillars enables the dissipation of spurious wave energy to the substrate, effectively mitigating unwanted spurious modes. Additionally, this novel approach involves anchoring the piezoelectric thin film to a supportive framework, consequently enhancing mechanical stability while simultaneously improving the heat dissipation capabilities of the core.

## 1. Introduction

The rapid advancement of global wireless radio frequency (RF) communication has driven the need for RF filters with higher operating frequencies and broader bandwidth capabilities. To address these growing demands, significant research efforts have been dedicated to investigating acoustic filters to ensure optimal communication quality within this congested spectrum. Currently, predominant commercial solutions include surface acoustic wave (SAW) resonators and thin-film bulk acoustic wave resonators (FBARs), which are known for their high performance and cost-effectiveness [1,2]. However, conventional SAW devices, fabricated using bulk lithium niobate (LiNbO_3_) or lithium tantalate (LiTaO_3_), encounter challenges in achieving ultra-high frequencies because of limitations on electrode width and the relatively low phase velocity (*v*) of acoustic modes propagating through the piezoelectric substrates [3,4]. While the resonance frequency (*f_s_*) of FBARs is primarily determined by their thickness, allowing for an increase in *f_s_* by merely reducing the thickness of the electrode and piezoelectric layer, this reduction in thickness inevitably leads to compromised deposition quality and increased ohmic loss [5,6]. Additionally, aluminum nitride (AlN)-based bulk acoustic wave (BAW) devices have faced constraints in achieving an effective electromechanical coupling coefficient ($K_{eff}^{2}$) greater than 7% [7,8], which is insufficient to meet the requirements of several designated 5G frequency bands.

In recent years, advances in film transfer technology have initiated a new era of research focused on thin-film-based acoustic resonators employing LiNbO_3_ or LiTaO_3_ materials. This groundbreaking approach has garnered significant interest and exploration [9,10,11,12,13,14,15]. Among the diverse cutting angles and acoustic modes, first-order antisymmetric (A1) Lamb mode resonators based on Z-cut LiNbO_3_ thin films offer a distinct advantage by simultaneously providing high *v* and high $K_{eff}^{2}$ value. These attributes considerably impact the central frequency and bandwidth of filters. Remarkably, seminal work by Yang et al. has demonstrated A1 mode resonators operating within the 5 GHz range, showcasing a $K_{eff}^{2}$ value of 29% [15], thereby highlighting their considerable potential in the sub-6 GHz domain. Nevertheless, a substantial challenge associated with A1 mode resonators is their vulnerability to spurious modes [16,17,18,19], which constitute a critical barrier to their extensive application. Consequently, the strategic goal of minimizing or eradicating these spurious modes becomes an essential consideration for the advancement of high-performance A1 mode resonators. An additional factor hindering the practical implementation of this configuration is its inherent restriction in power durability [20,21], because the piezoelectric plate is thermally isolated from the supporting substrate, leading to diminished thermal stability. Addressing these challenges will be crucial for realizing the full potential of A1 mode resonators in a variety of applications.

This paper presents a pioneering approach for A1 Lamb wave resonators based on LiNbO_3_ thin films by incorporating a support-pillar structure. Integration of the support structure facilitates dissipation of spurious acoustic wave energy to the underlying substrate, effectively mitigating the issue of multiple spurious modes. Moreover, this innovative approach involves anchoring the piezoelectric thin film with a robust support framework, thereby bolstering mechanical stability and enhancing the core’s heat dissipation capacity. Ultimately, these advances contribute to the improved performance and expanded applicability of antisymmetric Lamb wave resonators in practical scenarios.

## 2. Analysis of A1 and Spurious Modes

In this part, the properties of the A1 mode are systematically explored by assessing the propagation characteristics of A1 resonators with diverse geometries in a 600 nm Z-cut LiNbO_3_ thin film, employing finite-element analysis (FEA). To effectively stimulate the A1 mode in LiNbO_3_ thin films, suspended plates with top-only interdigital transducers (IDTs) are commonly employed, as illustrated in Figure 1. This schematic clearly delineates critical parameters, such as pitch (*P*), electrode width (*W*), electrode gap (*G*), and the thickness (*H*) of the LiNbO_3_ plate. By implementing periodic boundary conditions on both the left and right boundaries of the structures, the reflection effect caused by free edges can be significantly attenuated. Moreover, the IDTs positioned on the top surface exhibit alternating electrical connections, as indicated by the distinctive red and black highlights.

Initially, a frequency-domain analysis was undertaken to discern series resonance frequency (*f_s_*) and parallel resonance frequency (*f_p_*), employing the impedance spectrum of the A1 mode. Subsequently, the $K_{eff}^{2}$ value and 3 dB bandwidth quality factor (3 dB-*Q*) were ascertained through the application of the following equations:(1)$$K_{eff}^{2}=\frac{\pi^{2}}{4}\frac{f_{s}(f_{p}-f_{s})}{f_{p}^{2}}$$
(2)$$Q_{s}=\frac{\Delta f_{3dB}}{f_{s}}$$
(3)$$Q_{p}=\frac{\Delta f_{3dB}}{f_{p}}$$

Figure 2 presents the simulated *f_s_* and $K_{eff}^{2}$ as a function of IDT pitch (*P*). In terms of the $K_{eff}^{2}$ value, it is observed that an increased *P* results in a higher $K_{eff}^{2}$ for a given thickness. In contrast, *f_s_* demonstrates a significant decrease when *P* remains relatively small, ultimately stabilizing at approximately 3.01 GHz. This behavior can be understood, as *f_s_* is determined by the subsequent equation:(4)$$f_{s}=\sqrt{{\left(\frac{v_{z}}{2H}\right)}^{2}+{\left(\frac{v_{x}}{2P}\right)}^{2}}$$

In which *v_z_* and *v_x_* denote the phase velocities along the Z and X directions, respectively. As *P* expands to approximately 10 μm, the influence of further increasing *P* becomes negligible, causing *f_s_* to be predominantly determined by the thickness (*H*).

Subsequently, an eigenfrequency analysis was performed. In practical applications, high-performance devices generally confine acoustic energy within the main body by etching through the LiNbO_3_ thin film to create free boundaries. However, this energy confinement may lead to the emergence of undesirable spurious modes. To investigate the origins of these spurious modes in the resonator and devise strategies for their elimination, periodic boundary conditions ought not to be imposed on the simulation model. Instead, the model should closely mimic the actual device, as the number of IDTs can influence spurious modes, and periodic boundary conditions simulate an infinite number of electrode pairs. In this analysis, all sides of the resonator were assigned free conditions. Mechanical loss and dielectric loss factors were set to 0.002 to simulate the loss of the resonator under realistic conditions. Figure 3a illustrates the displacement mode shapes for *P* = 10 μm and a *W*:*G* ratio of 1:3. The A1 Lamb mode, a type of antisymmetric Lamb mode, displays displacement antisymmetry relative to the median plane of the wave. As observed in the figure, the majority of acoustic energy is concentrated between the electrodes, while a minor portion propagates to the area beneath the IDTs.

Essentially, the mechanical boundaries divide the resonator into two distinct sections. The first encompasses LiNbO_3_ sections without top electrodes, while the second contains LiNbO_3_ sections with top electrodes. As a result, the entire resonator body can be modeled as a cascading alternation of high- and low-impedance sections [17]. Acoustic waves undergo reflection at the interface or boundary between media with differing acoustic impedances. Since the A1 mode, confined between the IDTs, demonstrates the most significant mutual energy, it is regarded as the fundamental mode. In contrast, acoustic energy confined within the IDT region is classified as a spurious mode, as illustrated in Figure 3b. Due to the mechanical interfaces enabling internal reflections of acoustic waves, multiple orders of lateral spurious modes can be observed. As depicted in Figure 3c, in addition to the primary A1 resonance, several spurious modes are also excited within the resonator configuration. These spurious modes persist across the resonator’s bandwidth, exerting a negative impact on filter applications. 

## 3. Elimination of Spurious Modes with Supporting Pillars

Upon identifying the sources of spurious wave generation, we propose a novel structure to eliminate these spurious waves, as depicted in Figure 4a. In contrast to the conventional suspended-plate structure of the A1 resonator, the resonator’s main body in this design is supported by an array of pillars located beneath the electrodes. Depending on specific requirements, the support pillars can be configured as elongated or separated columns; for example, elongated pillars offer improved supporting strength and heat dissipation, while separated pillars facilitate the release of the sacrificial layer in the cavity during the device fabrication process. Nevertheless, to minimize acoustic loss, the pillars must be located directly under the top electrodes, with their width generally being less than or equal to that of the electrodes. In this paper, we set the support pillars to be elongated, the same as the electrodes structure, and their height is set to 2 μm. Figure 4b shows one of the feasible fabrication processes for the resonator with supporting pillars. The fabrication process begins with photolithography definition, followed by creating a patterned SiO_2_ embedded layer on the Si substrate using thermal oxidation treatment. Concurrently, a defect layer is formed within a specific depth on the bulk LiNbO_3_ substrate through He ion implantation. Next, the LiNbO_3_ substrate is bonded to the Si substrate with the SiO_2_ pattern, and an annealing treatment is applied. During this step, bubbles form in the defect layer, leading to the substrate’s separation and the transfer of the LiNbO_3_ thin film onto the Si substrate. Chemical mechanical polishing (CMP) is then employed to further control the thickness and reduce the surface roughness of the LiNbO_3_ thin film. Afterwards, IDTs are fabricated through e-beam and lift-off methods, ensuring that the electrode positions correspond with the Si pillar locations. The LiNbO_3_ thin film is etched using an inductively coupled plasma (ICP) technique to create release holes. Ultimately, vapor-HF (VHF) etching is carried out on the underlying SiO_2_ to form a cavity. It is true that the fabrication process for resonators with supporting pillar structures may be more challenging than that for the conventional suspended-plate structure resonators. The primary difficulty lies in the bonding step between the LiNbO_3_ and Si substrates, where certain areas of the Si substrate are thermally oxidized to form SiO_2_ as the sacrificial layer. In other aspects, the fabrication process is consistent with that of the suspended-plate structures and can be addressed using common MEMS technology. 

Figure 5 offers a comparative analysis of the spectral responses of resonators with Si pillar-supported and suspended-plate structures across various *P* values, in which the *W*:*G* ratio is fixed at 1:3 and the IDT number *N* = 2. As demonstrated in the figure, the resonators with pillar-supported structures display significantly fewer spurious modes in their spectral responses compared with those with suspended-plate structures. Furthermore, the presence of these pillars does not influence the resonant frequencies. Increasing electrode *P* or, equivalently, electrode *G* also contributes to the elimination of spurious waves. For resonators with pillar-supported structures, optimization can produce spectral responses devoid of any spurious modes. Based on the analysis of the A1 mode and spurious waves in the previous section, it becomes clear that the energy of the A1 mode is primarily focused between the region of the electrodes, while the energy of the spurious waves is mainly concentrated within the electrode-bearing area. The introduction of a pillar-supported structure assists in dissipating spurious waves, thereby aiding the elimination of spurious modes. Figure 6 depicts the intensity of normalized displacement across the regions along the midline of the resonator with *P* = 20 µm. It is shown that the displacement intensity primarily converges within the central region of the IDT. Furthermore, for the resonator with supporting pillars, the displacement within the IDT region undergoes a significant reduction. Increasing electrode *P* or *G* will increase energy concentration in the regions between electrodes, substantially reducing the vibration intensity in the electrode-bearing area. As a result, the primary mode is enhanced, leading to a reduction in spurious modes.

Subsequently, the *Q* values of these resonators are extracted, and Figure 7 presents a comparison of the *Q* values of the resonators with and without pillar-supported structures across different *P*. When the electrode periodicity or inter-electrode spacing is relatively small, a minor portion of the primary mode may leak into the substrate in addition to the spurious modes, causing the *Q* values of the resonator with a pillar-supported structure to be lower than that of the resonator with a suspended-plate structure. This effect is primarily reflected in the *Q_s_* values but can be mitigated by appropriately increasing the electrode *P*. As the electrode *P* increases, the *Q* values of the resonator with a pillar-supported structure converge with those of the resonator with a suspended-plate structure. Additionally, a moderate increase in the number of electrode pairs and optimization of the electrode structure can contribute to the enhancement of the resonator’s *Q* values. When the electrode period is fixed at 20 µm, in comparison to a resonator with only two IDTs, both the *Q_s_* and *Q_p_* values of the resonator with 20 electrodes exhibit a significant improvement. As for the *Q_s_* value, it increased from 108 to 287, while the *Q_p_* value rose from 397 to 460. This observation could be attributed to the fact that when the number of electrodes is relatively small, the total acoustic energy excited is also limited. Consequently, the device structure plays a significant role, with factors such as the distance between the electrodes and the edges, as well as the size of the supporting pillars, having a considerable impact. This could potentially result in an increased proportion of energy loss. Conversely, when the number of electrodes is larger, the total acoustic energy is greater and predominantly concentrated in the central region of the resonator. This leads to a reduced influence from the edge structure of the resonator, ultimately decreasing the proportion of energy loss in the acoustic wave. A similar result was also reported by [22].

## 4. Enhancing Power Capacity with Supporting Pillars

Comprehending the thermal conduction behavior of resonators and enhancing their power capacity is crucial for high-power filter applications [23,24,25,26]. Alterations in the ambient temperature within and surrounding the resonator may result in the filter failing to meet the desired performance. On the one hand, a significant rise in the device temperature caused by the self-heating effect can cause changes in the material’s elastic coefficient, leading to a shift in the resonant frequencies. On the other hand, a more critical issue is that the concentrated self-heating heat may induce film warping in the form of thermal stress. This warping can promote an increase in local current density, further exacerbating the self-heating effect and ultimately causing irreversible structural damage to the device.

In order to investigate the impact of pillars on the thermal behavior of resonators, the thermal transport properties of resonators with three distinct structures were simulated and compared, specifically those with Si pillars, SiO_2_ pillars, and a suspended structure. In all three configurations, the distance between the LiNbO_3_ plate and the Si substrate was set at 2 μm. In this simulation, heat conduction is considered the dominant effect, while convection is neglected due to the resonators’ relatively small size, and thermal radiation is not taken into account owing to the relatively low maximum temperatures. A fixed-temperature (293 K) boundary condition is applied at the back side of the Si substrate, which is also assumed to be the initial condition. In the lateral direction, a perfectly matched layer is applied to represent a large wafer. A surface heat source of 10^7^ W/m^2^ is applied to the top electrodes with a dimension of 10^−9^ m^2^. The temperature distribution diagrams of the resonators with Si pillars, SiO_2_ pillars, and suspended structures are displayed in Figure 8a, 8b, and 8c, respectively. As observed from the figures, the heat in the three structures is primarily concentrated in the effective central region of the resonator. The temperature increment of the Si pillar-supported structure is only about 2 °C, while the maximum self-heating temperature of the suspended structure reaches 423 K, indicating that Si pillars are advantageous for the device’s heat dissipation. Given that the thermal conductivity of SiO_2_ is lower than that of Si, the temperature rise of the structure with SiO_2_ support pillars is slightly higher than that of the Si support pillars, with a temperature rise of approximately 15 °C. However, this value is still considerably lower than the self-heating temperature rise of the suspended structure. LiNbO_3_ has a large negative temperature coefficient of frequency (TCF) of −95 ppm/°C. Additionally, SiO_2_ can serve as a temperature compensation component, as it possesses a positive TCF of 85 ppm/°C, thereby reducing the influence of temperature changes on the resonator’s frequency [27,28,29,30,31].

## 5. Conclusions

In this study, we introduce an innovative A1 mode resonator design that incorporates a support-pillar structure, offering significant advancements in the field of resonator technology. The integration of these support pillars effectively facilitates the dissipation of spurious wave energy to the substrate, which in turn successfully mitigates unwanted spurious modes. Although inclusion of the support pillars might also result in a reduction in the main mode energy and a consequent decrease in the *Q* value, predominantly impacting *Q_s_*, this detrimental effect can be efficiently counterbalanced by increasing the electrode pitch *P* or gap *G*. Furthermore, the addition of support pillars substantially bolsters the mechanical stability and power handling capacity of the resonator. Compared with the suspended-plate structure resonator, temperature increase attributed to the self-heating effect of the resonator featuring the support-pillar structure is markedly decreased, which is highly beneficial for enhancing overall power capacity. These advantages highlight the considerable potential of the proposed innovative structures, paving the way for further advancements and optimizations in resonator design.

## Figures and Tables

**Figure 1 micromachines-15-00195-f001:**
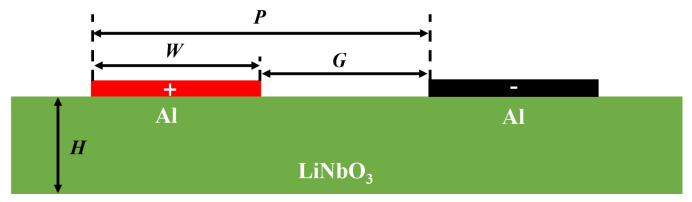
Illustration of the suspended LiNbO_3_ thin-film-based A1 resonator.

**Figure 2 micromachines-15-00195-f002:**
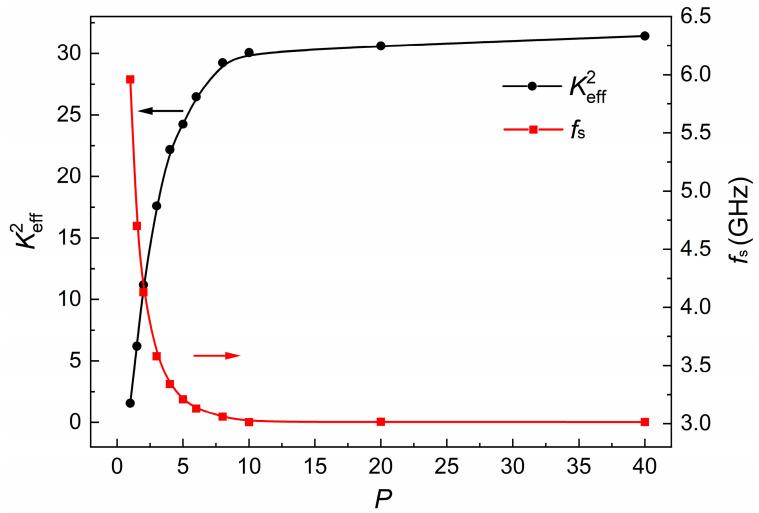
Simulated *f_s_* and $K_{eff}^{2}$ of A1 mode for different *P*; *H* is fixed at 600 nm.

**Figure 3 micromachines-15-00195-f003:**
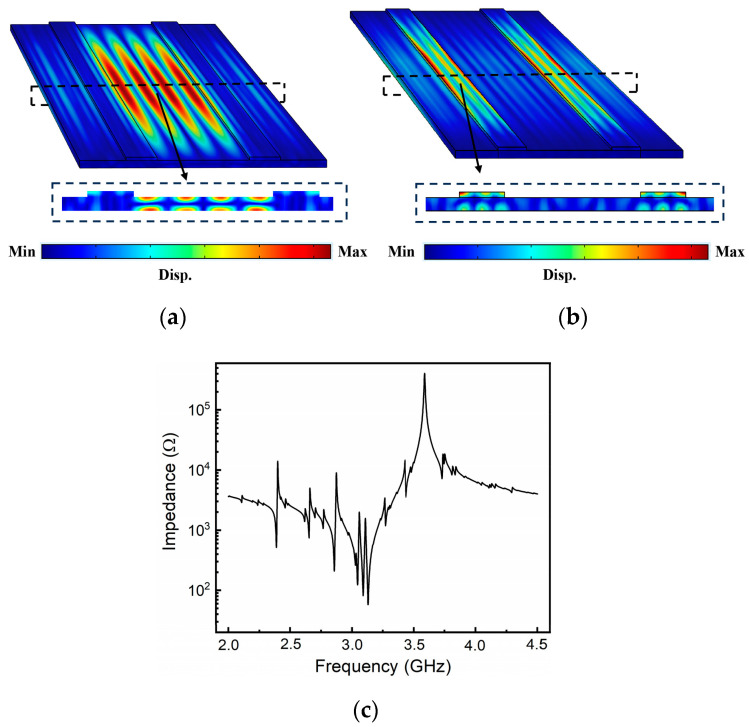
Simulated displacement mode shapes of (**a**) A1 mode (**b**) spurious modes, (**c**) the simulated spectral response with spurious modes.

**Figure 4 micromachines-15-00195-f004:**
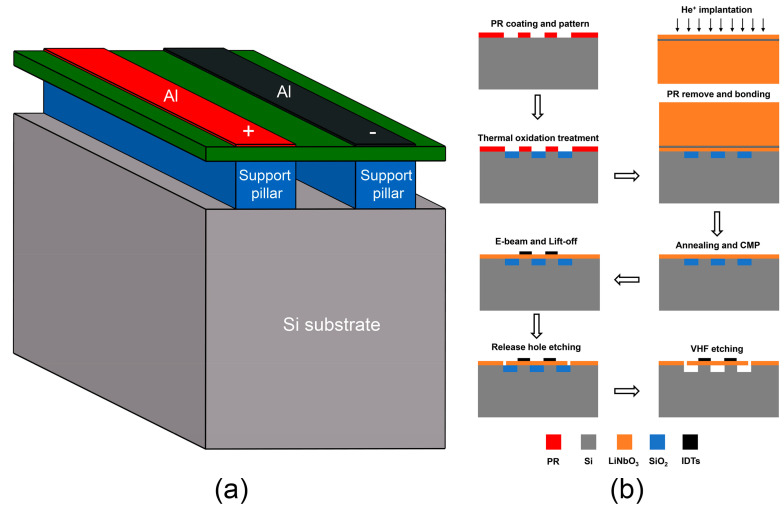
(**a**) Schematic diagram of the new type A1 resonator with support pillars, (**b**) A feasible fabrication process of the proposed new type A1 resonator.

**Figure 5 micromachines-15-00195-f005:**
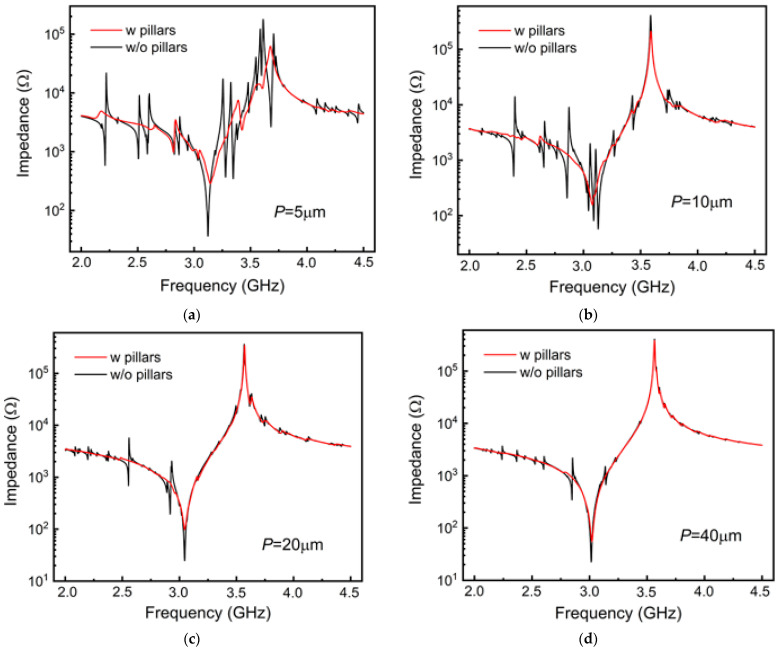
Comparison of the simulated spectral response between the pillar-supported resonators and the suspension resonators with (**a**) *P* = 5 μm (**b**) *P* = 10 μm (**c**) *P* = 20 μm (**d**) *P* = 40 μm (**e**) *P* = 60 μm (**f**) *P* = 80 μm.

**Figure 6 micromachines-15-00195-f006:**
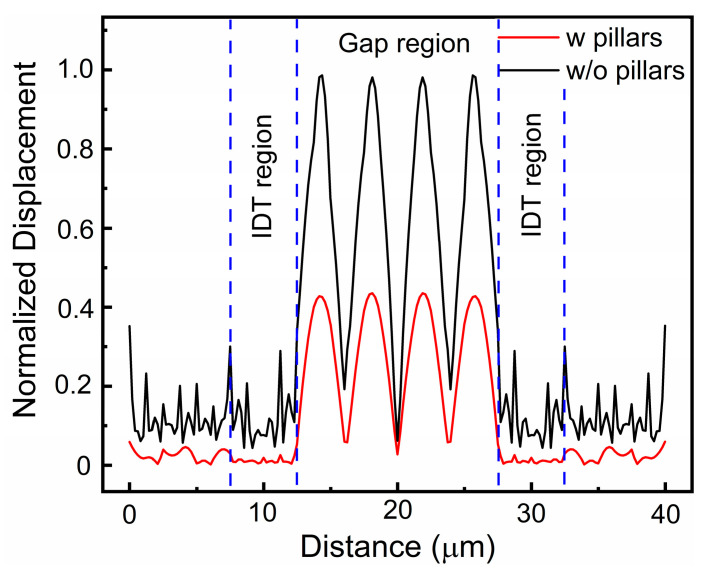
Comparison of the normalized displacement along the midline between the pillar-supported resonators and the suspension resonators.

**Figure 7 micromachines-15-00195-f007:**
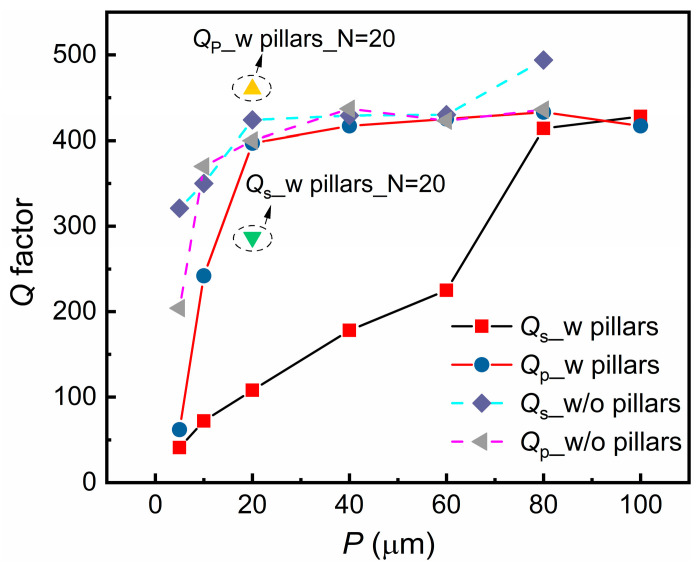
Extracted *Q* factors of the pillar-supported resonators and the suspension resonators with different *P*.

**Figure 8 micromachines-15-00195-f008:**
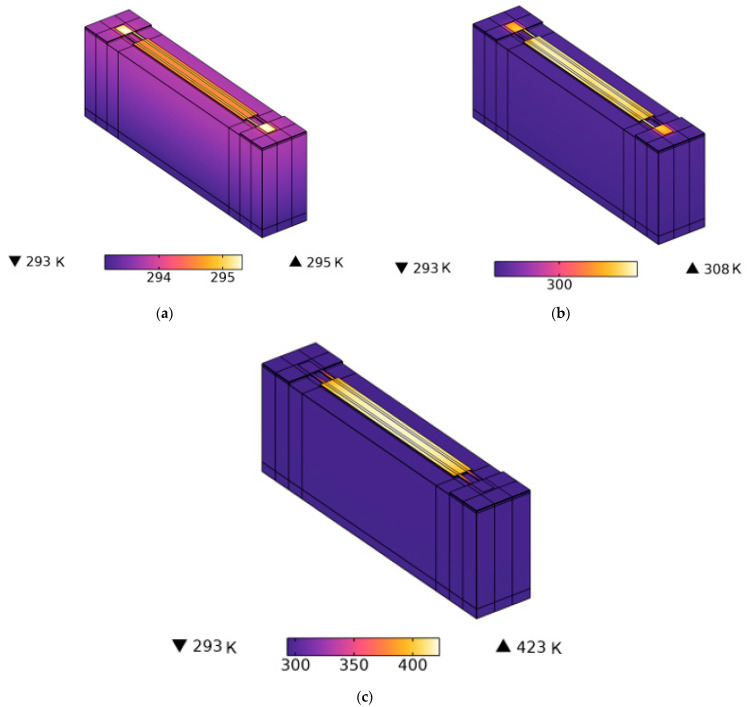
Simulated temperature distribution of (**a**) Si pillar-supported resonator (**b**) SiO_2_ pillar-supported resonator (**c**) suspended resonator.

## Data Availability

Data is contained within the article.

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
