# Peer review of "Lithium Niobate MEMS Antisymmetric Lamb Wave Resonators with Support Structures"

_micromachines, 2024, doi:10.3390/mi15020195_

Round 1
Reviewer 1 Report
Comments and Suggestions for Authors
This work presents a way of supporting pillar structures to decrease spurious modes and optimize heat dissipation in Antisymmetric Lamb Wave Resonators. This is an interesting paper, well organized, and suggested for publication after minor revision. Following, are the few concerns related to technical content that need to be addressed, and the paper should be carefully proofread for typos and grammar errors.
1 In line 20, the authors use spurious mode. I think it’s best to use spurious modes. Check it all around the paper and fix it.
2 In Fig 4, the quality needs to be improved. You could adjust the ratio of height to make it more suitable. More importantly, what is the structure (Size and shape) of the pillars?
3 In Fig 6, the values of the y-axis normalized displacement are not suitable. Check and fix it.
4 In lines 165~169, how do you prove them? Adjust the words, and give the evidence or citations.
5 In lines 213~215, how do you prove them? Adjust the words, and give the evidence or citations.
6 In Fig 8, there are too many blanks. You could place them in one line.
7 In lines 138-140, the authors claim the proposed resonators w/ pillar has significantly fewer spurious modes as compared to those w/o structures. However, in Fig. 5, more resonate mode is found in w/ pillar design, can the author explain why?
8 Do the authors have the experimental results to support this design, and what are the main pros and cons for the sensor with w/ pillar design, as compared to the conventional suspended one?
Comments on the Quality of English LanguageIt is generally good but need to be aware of the typos.
Reviewer 2 Report
Comments and Suggestions for Authors
This paper studies the LiNbO3 MEMS Resonator and proposes a new way to suppress spurious modes. The article is well written, the introduction gives a good overview of the problem. It discusses an acoustic resonator based on a very thin LiNbO3 plate or film (h = 600 nm), which operates in the A1 acoustic mode in a gap between electrodes several μm wide. To suppress parasitic acoustic vibrations, it is proposed to use support pillars on the back side of the film beneath the electrodes. It is shown that their presence significantly suppresses parasitic oscillations and a reasonable mechanism for this process is proposed. However, this is purely theoretical work and the technology for creating such a resonator has not been considered at all. Therefore, the shortcomings of the article include the following:
1. A suspended thin film resonator without any substrate was considered. Of course, such a resonator can be simulated, but the experimental implementation of such a resonator will cause great difficulties, since usually a thin piezoelectric film is formed on some material substrate.
2. The presence of support pillars on a silicon substrate assumes that an air cavity without any supporting electrode will be located between the working area of the resonator and the substrate, which is also difficult to imagine in practice.
3. The results of section 4 can be easily explained by the fact that suspended structure is considered. Heat removal from it will be difficult because the thermal conductivity of air is thousands of times less than the thermal conductivity of silicon. If the entire surface of the resonator were located on silicon, its temperature would be even lower.
4. Figure 6 shows a Normalized displacement distribution that reaches 15. However, normalization assumes that the maximum value of the normalized value should be equal to one.
Thus, the work can be recommended for publication if the authors explicitly indicate that exclusively theoretical results were obtained or add references to works that consider suspended thin film resonators made of LiNbO3 of such thickness.
